# CLEEGN: A Convolutional Neural Network for Plug-and-Play Automatic EEG Reconstruction

## Abstract

Human electroencephalography (EEG) is a brain monitoring modality that senses cortical neuroelectrophysiological activity in high-temporal resolution. One of the greatest challenges posed in applications of EEG is the unstable signal quality susceptible to inevitable artifacts during recordings. To date, most existing techniques for EEG artifact removal and reconstruction are applicable to offline analysis solely, or require individualized training data to facilitate online reconstruction. We have proposed CLEEGN, a novel convolutional neural network for plug-and-play automatic EEG reconstruction. CLEEGN is based on a subject-independent pre-trained model using existing data and can operate on a new user without any further calibration. The performance of CLEEGN was validated using multiple evaluations including waveform observation, reconstruction error assessment, and decoding accuracy on well-studied labeled datasets. The results of simulated online validation suggest that, even without any calibration, CLEEGN can largely preserve inherent brain activity and outperforms leading online/offline artifact removal methods in the decoding accuracy of reconstructed EEG data. In addition, visualization of model parameters and latent features exhibit the model behavior and reveal explainable insights related to existing knowledge of neuroscience. We foresee pervasive applications of CLEEGN in prospective works of online plug-and-play EEG decoding and analysis.

## 1 Introduction

Since the first record of human electroencephalogram (EEG) performed almost a century ago (in 1924), EEG has been one of the most widely used non-invasive neural modalities that monitors brain activity in high temporal resolution (Koike et al., 2013; Mehta & Parasuraman, 2013; Sejnowski et al., 2014). Among a variety of modalities, EEG has extensive use in the clinical assessment of neurological and psychiatric conditions, as well as in the research of neuroscience, cognitive science, psychology, and brain-computer interfacing (BCI).

EEG signals measure subtle fluctuations of the electrical field driven by local neuroelectrophysiological activity of a population of neurons in the brain cortex (Cohen, 2017). While the electrodes are placed on the surface of the scalp, undesired artifacts may introduce interruption in the measurements and distort the signal of interest. Even in a well-controlled laboratory with a well-trained subject who can maximally keep the body still and relaxed, the EEG signals, unfortunately, could be contaminated by inevitable behavioral and physiological artifacts such as eye blinks, reflective muscle movements, ocular activity, cardiac activity, etc (Croft & Barry, 2000; Wallstrom et al., 2004; Romero et al., 2008). In practice, it is difficult to identify and track the sources of artifacts entirely due to their diversity and non-stationarity. Noise cancellation and artifact removal remain major issues in EEG signal processing and decoding.

Currently, numerous methods have been proposed to alleviate the influence of artifacts in EEG signals. Traditional EEG denoising algorithms include filtering, regression, data separation or decomposition (Makeig et al., 1995; Islam et al., 2016; Kothe & Jung, 2016). According to previous meta-analyses on EEG artifact removal literature (Urigüen & Garcia-Zapirain, 2015; Jiang et al., 2019), independent component analysis (ICA) is especially popular. It is majorly used in 45% of

EEG denoising literature. ICA-based artifact removal estimates the component activity by unmixing the EEG data in the channel domain. Through manual or automatic identification, one can exclude the artifact components and then reconstruct the EEG data through back projection based on non-artifact components (Jung et al., 2000a). The fast growth of deep learning methods has drawn state-of-the-art performances in a variety of machine learning problems (LeCun et al., 2015). Lately, deep-learning-based EEG reconstruction has drawn much attention (Leite et al., 2018; Sun et al., 2020; Lopes et al., 2021; Lee et al., 2020; Chuang et al., 2022). Although these methods can effectively remove artifacts from artificial synthetic signals, their performance in reconstructing real EEG data has not yet been validated in terms of decoding labeled EEG data. Meanwhile, the model design of existing deep-learning-based techniques for EEG reconstruction rarely takes the characteristics of EEG into account.

In this work, we propose CLEEGN, a ConvoLutional neural network for EEG reconstructioN. CLEEGN is capable of subject-independent EEG construction without any training/calibration for a new subject. The contributions of this work are three-fold:

- a light-weight autoencoder CNN, CLEEGN, with a subject-independent framework that facilitates plug-and-play EEG reconstruction.
- CLEEGN outperforms leading online/offline methods in providing reconstructed EEG data with the best decoding performance for BCI datasets.
- with a novel model design dedicated to EEG reconstruction, CLEEGN characterizes patterns of EEG interpretable and provides neuroscientific insights.

## 2 RELATED WORK

Current processing techniques for EEG artifact removal are highly varied based on the context where the algorithm may apply. Earlier attempts of EEG denoising assumed that the EEG signals and artifacts appear in different frequency ranges. Based on the assumption, some significant artifacts can be eliminated by the linear filtering method during the online stage (Seifzadeh et al., 2014). Despite the advantages of low computational time, linear filtering hardly removes artifacts that distribute in an overlapped frequency range of EEG signals. Another approach, adaptive filtering (Schlögl et al., 2007), estimates artifact signals through additional EOG, EMG, ECG channels and removes these noisy signals from the recording signals by regression. Nevertheless, this approach requires additional auxiliary electrodes and raises the cost and inconvenience in practical applications. The blind source separation (BSS) method in EEG denoising was developed by assuming that the recording EEG signals are linear combinations of the signals from noise sources and the brain neurons. One of the most well-known BSS method is independent component analysis (ICA) (Jung et al., 2000a;b), which is able to separate EEG signals into independent components (ICs) (Makeig et al., 1995).

Traditionally, the artifact components extracted by ICA are determined and removed through manual inspection. Recently developed ICLabel can label the ICs provenance into seven different categories: brain, eye, heart, muscle, line noise, channel noise, and other (Pion-Tonachini et al., 2019). Artifact subspace reconstruction (ASR) is another automatic approach, which is based on the principal component analysis (PCA) method (Kothe & Jung, 2016). The ASR method selects relatively noiseless periods from the multi-channel EEG data as reference based on the data distribution. After projecting all EEG data to the principal-component domain, high-variance components projected from the artifacts are detected by a cutoff parameter $k$. The noiseless signals are reconstructed by preserving the components without carrying artifacts and back-projected to the time domain. The ASR method has been shown capable of improving the quality of ICA decomposition (Chang et al., 2020).

Recently, neural network-based methods have been proposed to remove artifacts for EEG data. A variety of network structures have been applied to the framework for removing EEG artifacts and reconstructing clean EEG. A deep convolutional autoencoder (Leite et al., 2018) can enhance the peak-signal-to-noise ratio compared to the linear filtering method via a common CNN autoencoder structure, which has been widely used on image denoising. Their work shows that it seems practicable to transform the EEG waveform through a CNN structure. Later on, a combined framework integrating Bayesian deep learning and ICA (Lee et al., 2020) used thresholding of the EEG data distribution to discard ICs classified as ocular artifacts. These methods leverage the flexibility of deep learning model design and achieve improvements in their assessments. Considering the non-

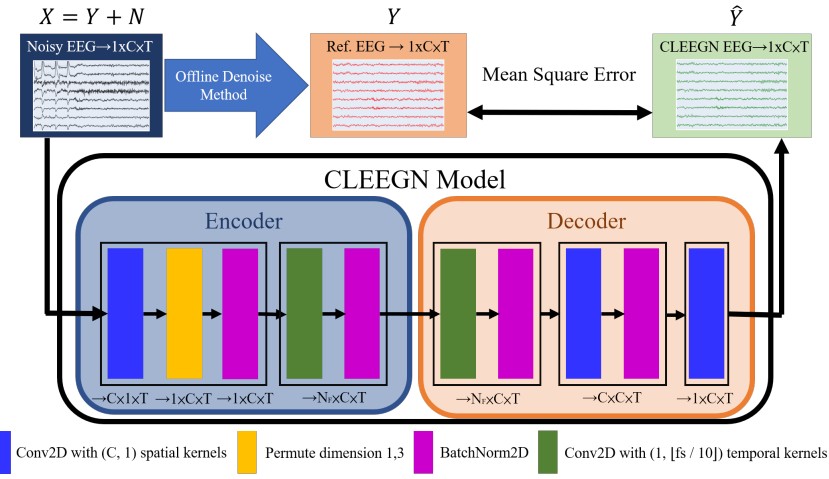

Figure 1: Illustration of the proposed CLEEGN model architecture and the model training flow.

stationary property in EEG data and the degradation phenomenon during training, 1D-ResCNN (Sun et al., 2020) was proposed, which adopted an Inception-Residual module in the network structure. This network is able to remove EOG, ECG, EMG on single-channel synthesis EEG data. Instead of using synthesis EEG data, IC-U-Net (Chuang et al., 2022) was created, which generated pairings of noisy and noiseless EEG data through ICA as training data. The proposed neural network is a one-dimensional adaptation of U-Net architecture trained with the ensemble of four loss functions to minimize the difference between amplitude, velocity, acceleration, and frequency components from the signals. This work showed that their reconstructed signal has higher SNR and can surely increase the number of brain components classified by ICLabel.

## 3 MATERIALS AND METHODS

**CLEEGN Architecture and Model Training.** CLEEGN is designed to map multi-channel noisy EEG into a latent space and reconstruct it into noiseless EEG signals. The architecture of CLEEGN is shown in Figure 1. The architecture of the encoder is inspired by an existing CNN model designed for EEG recognition (Wei et al., 2019) and incorporates convolutional blocks that capture spatiotemporal characteristics of EEG data. The first convolution block is used to extract spatial EEG features through a convolutional layer containing $C$ spatial filters whose shape is $(C, 1)$, where $C$ is the number of electrodes of the EEG signals. The second convolution block is used to extract temporal EEG features through a convolutional layer containing $N_F$ temporal filters with shape $(1, \lfloor f_s \times 0.1 \rfloor)$,

Table 1: The architecture of CLEEGN.

| Block | Layer | #kernels | Size | Output shape |
|---|---|---|---|---|
| Encoder | Input | | | $(B, 1, C, T)$ |
| | Conv2D | $C$ | $(C, 1)$ | $(B, C, 1, T)$ |
| | Permute | | | $(B, 1, C, T)$ |
| | BatchNorm | | | |
| | Conv2D | $N_F$ | $(1, \lfloor f_s/10 \rfloor)$ | $(B, N_F, C, T)$ |
| | BatchNorm | | | |
| Decoder | Conv2D | $N_F$ | $(1, \lfloor f_s/10 \rfloor)$ | $(B, N_F, C, T)$ |
| | BatchNorm | | | |
| | Conv2D | $C$ | $(C, 1)$ | $(B, C, C, T)$ |
| | BatchNorm | | | |
| | Conv2D | $1$ | $(C, 1)$ | $(B, 1, C, T)$ |

$C$: # channels, $T$: # time points, $f_s$: Sampling rate, $B$: Batch size

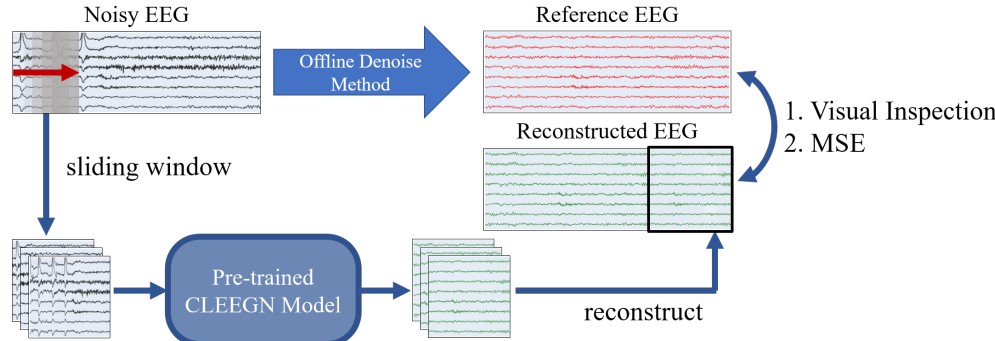

Figure 2: Schematic of the simulated online processing flow and the assessments of fitness between the reconstructed and reference EEG signals.

where $f_s$ denotes the sample frequency of the EEG signals. As for the decoder, we design an approximately symmetric structure to the encoder with three convolutional blocks. The first convolutional block decodes the EEG feature with $N_F$ temporal kernels with shape is $(1, \lfloor f_s \times 0.1 \rfloor)$. The second block decodes the EEG feature by a convolution layer with $C$ spatial kernels of shape $(C, 1)$. The last convolutional block is for projecting the feature domain back to the original time domain. To maintain the shape between the model inputs and outputs, every convolution block applies zero-padding except the first block in the encoder. The detail of the CLEEGN architecture is available in Table 1.

As illustrated in Figure 1, the objective of the proposed method is to minimize the difference between noiseless signals, $Y$, and the model output, $\hat{Y}$. The recording EEG data, $X$, is the combination of brain signals, $Y$, and the signals from multiple noise sources, $N$. CLEEGN can be regarded as a denoising autoencoder that is intended to perform artifact removal on EEG data by creating a mapping between multi-channel noisy EEG signals, $X$, and noiseless signals, $Y$. The training process of CLEEGN utilizes pairing noisy raw EEG data and noiseless reference EEG data so that the model can learn to transform noisy EEG data into reconstructed EEG data with maximal similarity to the reference data. To generate large-scale and real reference EEG data, we adopted automatic denoising methods, ICLabel and ASR, to remove artifacts and reconstruct clean waveforms offline. Rather than synthetic data, the use of real EEG data ensures presence of artifact/noise in a natural way and thus provides a realistic evaluation for our model. The pairing EEG recording with $C$ channels of continuous noisy/noiseless data was further segmented into training sample pairs. The input size is $(C, T)$, where $T$ is the number of time points based on the sampling rate of the EEG data in 4-second segments. The windows size and stride are set as $T$ and $0.5T$. Considering the context of plug-and-play EEG reconstruction, we perform subject-independent training scheme where EEG data into $k$ disjoint sets by subjects. During the training process, one of the sets was left out for testing. Subjects' EEG data in the left-alone set were not involved in both training and validation. A complete experiment on a single dataset would result in $k$ different models. The artifact removal performance of a model was evaluated by using the left-alone set. The number of subjects in one set and the EEG duration available for each subject depend on the experiment setting and the dataset used.

**EEG Reconstruction and Evaluation.** The reconstructed EEG data of CLEEGN is generated through an online reconstruction simulation. In the online stage, the system performs artifact removal on a new subject using a pre-trained CLEEGN model without any requirement of training/calibration. Figure 2 shows the online EEG reconstruction simulated in offline. The EEG data from a subject in the left-alone set would be fed into the model sequentially. The stride size of the sliding window is set to $0.5$ seconds to minimize the delay in online reconstruction.

The fitness of the model can be evaluated by observing the waveform visually or measuring the similarity using the mean square error (MSE) between reconstructed and noiseless EEG. Meanwhile, we propose to use the decoding performance of labeled EEG data as an objective measurement of the reconstructed EEG quality. We employed EEGNet, a compact CNN for end-to-end EEG decoding (Lawhern et al., 2018), as the classifier to decode the labeled EEG data in our study.

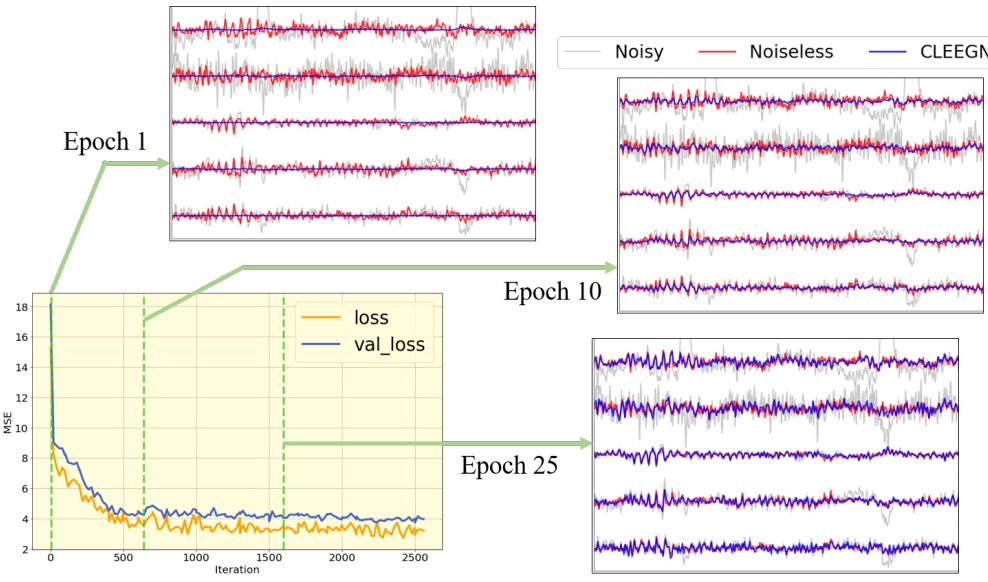

Figure 4: Reconstructed EEG signals by CLEEGN across training steps.

# 4 EXPERIMENTS

**Data and Model Fitting.** In this work, we applied two labeled EEG datasets, the BCI-ERN dataset [1] and the MAMEM-SSVEP-II dataset [2]. We estimate the classification performance by calculating the score of Area Under the Curve (AUC) in ERN binary classification and Top-1 accuracy in the SSVEP task. Figure 3 is the schematic diagram of the process of decoding performance estimation using the reconstructed EEG data. The segmented EEG epochs are divided into three splits within each subject: training set, validation set, and test set. The ratio between classes remained the same in each set. We perform 20 repeated runs with

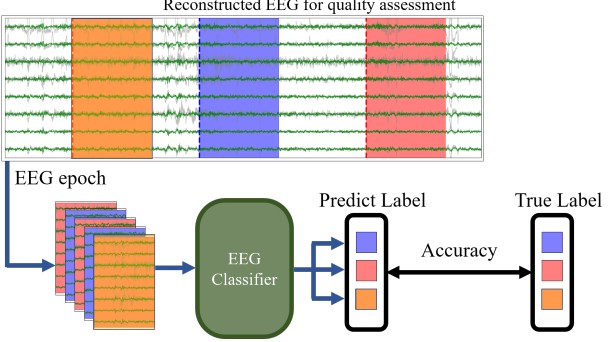

Figure 3: Objective EEG quality assessment of the reconstructed EEG based on the decoding performance.

shuffled data to estimate the average performance of each decoding performance evaluation. Detailed descriptions of the two datasets are available in the appendix at A.2. Figure 4 presents the loss curve and the reconstruction result at the end of three different training iterations in a sample run using the ERN dataset. Through this exploration, we can look into the course of model fitting. In the early stage of training, the model reduces the large MSE loss by smoothing the input signals. To further minimize the error between reconstruction and reference (offline denoised signal), the model compensates the intensity to better fit the reference signals. At the convergent stage of training, we can see the reconstructed waveform is very similar to the one in reference data.

**Types of Reference Data.** As ICLabel, ASR, and the hybrid ASR-ICLabel are employed to automatically generate large-scale noiseless reference EEG data offline, it is of our interest to investigate what type of reference EEG serves as the best noiseless reference data for the CLEEGN model training. Figure 5 presents the EEG waveforms sampled from Subject 2 in the ERN dataset. The

---

[1]https://www.kaggle.com/c/inria-bci-challenge
[2]https://www.mamem.eu/results/datasets/

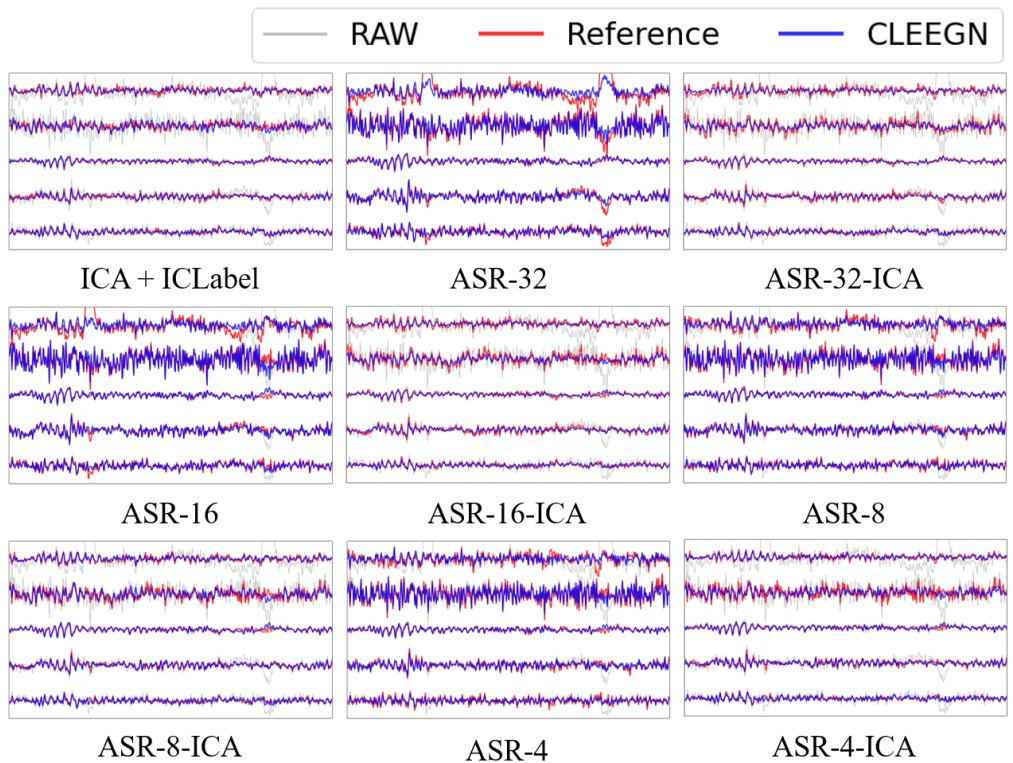

Figure 5: Visualization of raw (gray), reference (red), CLEEGN (blue) EEG waveforms with offline methods by ICLabel, ASR-32, ASR-32-ICLabel, ASR-16, ASR-16-ICLabel, ASR-8, ASR-8-ICLabel, ASR-4, ASR-4-ICLabel. Each segment plots a five-second segment of signals at Fp1, T7, Cz, T8, and O2.

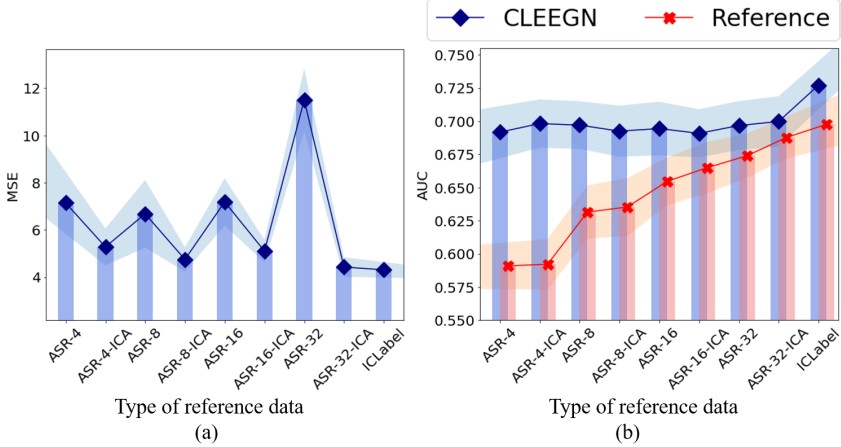

Figure 6: (a) Overall fitness of the CLEEGN model across types of reference data using the ERN dataset. (b) Decoding performance of the CLEEGN-reconstructed EEG data (blue) and the corresponding reference data used for CLEEGN model training (red).

EEG time series from top to bottom represent the recording of Fp1, T7, Cz, T8, and O2 channels. The nine sub-figures visualize the comparison between the noisy raw data and the reference data reconstructed by the ICLabel, ASR with or without ICLabel when the cutoff parameter $k$ was set to 4, 8, 16, and 32. We can see the waveform of ICLabel and the ASR-ICLabel hybrid method with

4 different cutoff parameters $k$ are similar, and the noiseless waveform reconstructed by CLEEGN are highly correlated to the result from these 5 methods. High-amplitude ocular artifacts such as blinking and eye saccades are phenomenal in frontal channels (Fp1) and muscular artifacts featuring high-frequency (around 24-30 Hz in the $\beta$ band) are observed on the T7 channel. CLEEGN, ICA, and ASR-ICLabel hybrid methods can eliminate these kinds of artifacts. Compared to the ICLabel and ASR-ICLabel hybrid methods, ASR is unable to eliminate the high-frequency EMG artifact for this dataset. The tolerance of large amplitude EOG artifact increases with the employment of larger cutoff parameter $k$. Limited by the reference data generated by ASR, CLEEGN preserves the high-frequency artifact. As for the large amplitude artifacts that failed to be identified by ASR-32 and ASR-16, we can see that CLEEGN is able to mitigate those EOG artifacts. As shown in Figure 6(a), the reference data prepared by ICLabel offer the lowest MSE, i.e. the best fitness for CLEEGN training. We also compare the types of reference EEG data and their corresponding CLEEGN reconstruction results in the decoding performance of the ERN EEG dataset to assess their data quality. As illustrated in Figure 6(b), CLEEGN reconstruction draws better performances for all denoising methods. Although the performance decreases with a smaller cutoff parameter $k$, CLEEGN can outperform the reference data used for its training. This result suggests that our method not only removes the artifact but also preserves informative brain activity in the EEG under our cross-subject training scheme. As the ICLabel provides the reference data with the best decoding performance and fitness, we chose ICLabel as the source of reference data for further experiments in this study.

**Training Data Size.** We explored the effect of data length per subject in the training set regarding the fitness and the decoding performance of CLEEGN. The training data were segmented from the first 1, 2, 4, 10, 20, and 30 minutes of data in each EEG recording to investigate the trade-off between training time and reconstruction performance. Figure 7(a) shows the fitness of CLEEGN to the reference data. The model trained using the first 10 minutes obtain the minimal value among all duration configurations. Though the difference in MSE between 10, 20, and 30 minutes is not noticeable, Figure 7(b) shows that using the 10-minute training data yields the best decoding performance among all settings. Interestingly, with only one-minute training data from each subject, CLEEGN can achieve comparable decoding performance to the reference data. This indicates that CLEEGN retains its performance even when each subject only contributes a short recording for training. In addition, we explore the effect of the number of subjects included in the training set on the fitness of CLEEGN model training and the decoding performance of the reconstructed EEG data. We randomly reduced the number of subjects for training from 12 to 6, 4, and 2. Since the subset of subjects may influence the performance, we tested multiple combinations and averaged

Table 2: Overall performance over all subjects in the BCI-ERN dataset.

| Method | MSE ↓ | AUC ↑ | Total parameters ↓ |
|---|---|---|---|
| ICLabel (reference) | – | 0.7218±0.0197 | – |
| Raw data (noisy) | 65.0345±12.5794 | 0.5578±0.0100 | – |
| 1D-ResCNN | 6.7147±0.5025 | 0.6697±0.0175 | 325891 |
| RNN | 10.6686±0.9257 | 0.6708±0.0180 | 787984 |
| SCNN | 7.7823±0.6403 | 0.6750±0.0172 | 16815552 |
| IC-U-Net | 5.2086±0.4204 | 0.6949±0.0140 | 2683192 |
| CLEEGN | **3.5984±0.2538** | **0.7252±0.0189** | **220755** |

Table 3: Overall performance over all subjects in the MAMEM-SSVEP-II dataset.

| Method | MSE ↓ | Accuracy ↑ | Total parameters ↓ |
|---|---|---|---|
| ICLabel (reference) | – | 0.4940±0.0667 | – |
| Raw data (noisy) | 1.3692±1.0449 | 0.2782±0.0058 | – |
| 1D-ResCNN | 0.2047±0.0524 | 0.2908±0.0316 | 313735 |
| RNN | 0.1097±0.0151 | 0.2806±0.0198 | 751516 |
| SCNN | 0.1002±0.0134 | 0.2959±0.0264 | 16038324 |
| IC-U-Net | **0.0698±0.0085** | 0.3422±0.0253 | 2664724 |
| CLEEGN | 0.0777±0.0111 | **0.5159±0.0622** | **14043** |

the results. With the decrease in the number of subjects, the MSE value and the standard error (light span area) increase in Figure 8(a), which indicates that the generalization ability of the CLEEGN model reduces when fewer subjects are included for training. In Figure 8(b), we can observe a slight decrease in decoding performance. We consider the number of subjects as an essential factor in the performance of CLEEGN, yet it requires only a few subjects to achieve a satisfactory performance compared to the reference data.

**Performance Comparison.** We compare the performance of CLEEGN against five baseline methods, ICLabel, 1D-ResCNN (Sun et al., 2020), IC-U-Net (Chuang et al., 2022) and the simple CNN (SCNN) structure and RNN structure proposed in a previous work (Zhang et al., 2020). Except for the ICLabel that operates offline, CLEEGN and the other neural network-based methods perform a simulated online reconstruction based on the same training process with the reference data generated by ICLabel. Results of the ERN EEG dataset are shown in Table 2. CLEEGN has the best fitness to the reference data with the

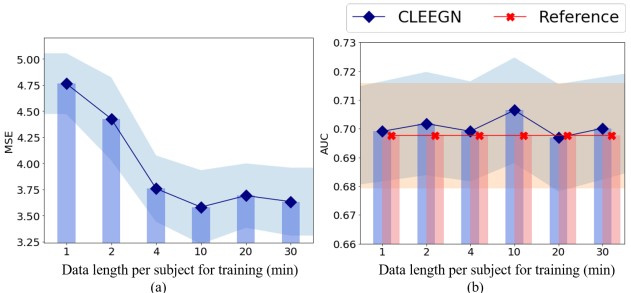

Figure 7: Performance of CLEEGN against the training data length per subject evaluated by the BCI-ERN dataset on (a) the fitness to the reference data; and (b) the decoding performance.

minimum MSE, the highest AUC score in the decoding performance, and the least parameters. For the SSVEP EEG data, although not having the minimum MSE, CLEEGN outperforms other methods in the decoding accuracy and the model size. The evaluation across the two datasets suggests an overall superiority of our proposed CLEEGN model over other existing neural network-based methods in online reconstruction. CLEEGN even provides a better reconstruction than the offline ICLabel. These promising results indicate the usability of CLEEGN in online training/calibration-free EEG reconstruction that truly meets the need for real-world applications of EEG-based BCI.

**Visualization.** On account of the interpretability of the CLEEGN model, we visualized its intermediate latent features by mapping onto a 2-dimensional domain based on the principal component analysis (PCA) of the noisy raw EEG data (Wold et al., 1987). Figure 9(a) shows the principal component space and the noisy EEG data channel projections. We observe that the scatters of noisy EEG data retain the spatial relationship of the actual EEG electrodes. The arrangement of frontal electrodes (prefix in Fp, P) to the posterior (prefix in P, O) are projected along the x-axis from right to left. The projection from top to bottom matches the left-side electrodes (suffixed in odd num-

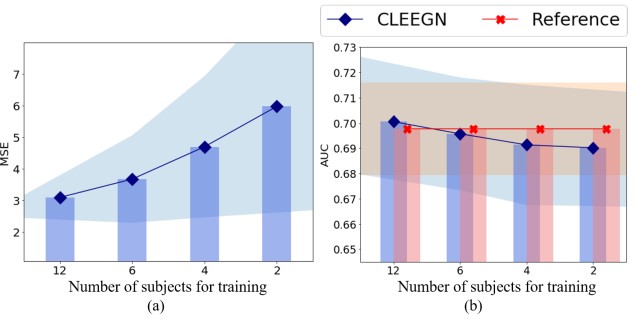

Figure 8: EEG reconstruction performance of CLEEGN against the number of subjects evaluated under the BCI-ERN dataset by (a) the fitness to the reference data; and (b) the decoding performance.

bers), central electrode (suffixed in z), and right-side electrodes (suffixed in even numbers). The projection suggests that EEG data of adjacent channels tend to show similar waveforms. The blue dots in Figure 9(b), (c) are the projection of latent features resulting from the first and second convolutional layers, which make up the encoder design of CLEEGN. The projection of latent features from the next two convolutional layers is presented in Figure 9(d), (e). From Figure 9(b) to (e), we can see the distribution range of the latent features shrink in the encoder design and expand in the decoder design of CLEEGN. Figure 9(f) compares the distribution of noisy data and CLEEGN

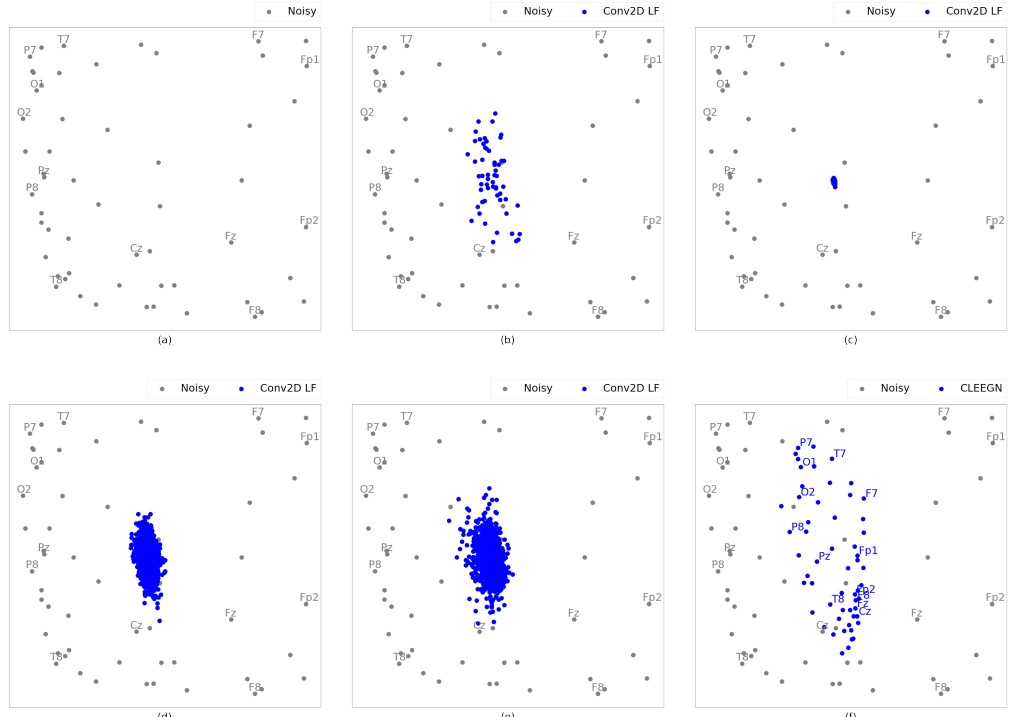

Figure 9: PCA visualization of latent features in CLEEGN for a single subject in the ERN dataset. (a) The noisy EEG data projected on the PCA space of the noisy EEG data. (b)-(e) Latent features in the first to fourth convolutional layers projected on the original noisy PCA space. (f) CLEEGN reconstructed EEG data projected on the original noisy PCA space.

reconstructed data, showing that the CLEEGN reconstructed data is a more compact cluster than the noisy EEG in the PCA space. This implies that the process within CLEEGN includes a projection of the original EEG data in the upstream layer, complex temporal filtering and combination in the midstream layers, and a final projection that converts the noiseless latent features back to the channel domain.

## 5 CONCLUSION

In this work, we have proposed CLEEGN, a novel convolutional neural network for plug-and-play automatic EEG reconstruction. The training of CLEEGN leverages the conventional offline denoising methods, ASR and ICLabel, with automatic component classification to generate abundant noiseless EEG data. The performance of CLEEGN was objectively validated using multiple evaluations including waveform observation; reconstruction error assessment; and decoding accuracy on well-studied, labeled datasets. The experiment results suggest that, even without any calibration, CLEEGN can predominantly preserve inherent brain activity. According to the decoding performance of reconstructed EEG data, CLEEGN outperforms other neural network-based denoising methods on both ERN and SSVEP EEG decoding. From the visualization of the waveform, CLEEGN can remove artifacts from different sources and the waveform is highly correlated to the reference data. Through the visualization of model parameters and latent features, we exhibit model behavior and reveal explainable insights related to existing knowledge of neuroscience. CLEEGN effectively learns the transformation of EEG reconstruction from existing technique and even outperforms conventional offline approach. Future extension of this work includes incorporating other EEG denoise methods and their mixed use, and enhancing the inference ability across datasets or even across recording montages. We foresee pervasive applications of CLEEGN in prospective works of EEG decoding and analysis.

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

# A  APPENDIX

## A.1  NEURAL NETWORK TRAINING CONFIGURATION

### A.1.1  CLEEGN

For training CLEEGN, we used the Adam optimizer with an initial learning rate of 1e-3 without weight decay. Besides, the exponential learning rate scheduler is applied with a gamma of 0.8. As for the loss function, we used Mean Squared Error (MSE). The batch size is set to 64 and the total training epoch is 40. During the training procedure, the model is evaluated using the validation subset at the end of every epoch with the purpose of saving the weights that achieved the lowest validation loss.

### A.1.2  IC-U-NET

The optimizer adopted in IC-U-Net (Chuang et al., 2022) training is SGD with an initial learning rate of 1e-2, momentum of 0.9, and weight decay of 5e-4. The learning rate scheduler used in the training procedure is the multistep scheduler. As for the loss function, a novel ensemble loss proposed in IC-U-Net is adopted. This ensemble is a simple linear combination of the Mean Squared Error (MSE) in amplitude, velocity, acceleration, and frequency components of EEG signals. Each term in the loss function has same weight. The batch size is set to 64 and the total training epoch is 200. The weight saving strategy in IC-U-Net training is the same as CLEEGN.

### A.1.3 1D-RESCNN

The Adam optimizer with an initial learning rate of 1e-3 and the Mean Squared Error (MSE) loss function is adopted in 1D-ResCNN (Sun et al., 2020) training. Since the 1D-ResCNN is developed under one-dimensional synthesized EEG data and no explicit instruction of using multi-channel EEG provided, we trained 1D-ResCNN using two different method. For one method, we trained multiple models for each EEG channel respectively. For the other method, we viewed each multi-channel EEG segment as a batch and the channel arrangement within a batch is fixed. Under our experiment, the result showed that the second method not only provides an efficient training process, but also results in a better reconstructed performance. The same weight saving strategy in CLEEGN is adopted in 1D-ResCNN training.

### A.1.4 EEGDENOISENET (SCNN, RNN)

Two further simple network architecture are adopted in the comparison, the simple convolutional structure and recurrent network structure. Mean Squared Error (MSE) loss function is adopted as the objective criterion. The weight saving strategy in SCNN and RNN training is the same as CLEEGN.

### A.1.5 DECODING PERFORMANCE EVALUATION MODEL: EEGNET

EEGNet (Lawhern et al., 2018) is a famous EEG decoding model and widely used in EEG literature. In the original EEGNet paper, they investigated their proposed model with a different number of kernels and denoted the model with $F_1$ temporal filters and $D$ spatial filters as EEGNet-$F_1$,$D$. We use two different settings to train the two datasets used. In ERN classification, we use the EEGNet-8,2 structure suggested by the EEGNet paper. As for the SSVEP classification, an experimental result showed that EEGNet-100,8 can draw the best performance. We trained and evaluated the decoding performance individually for each subject. We divided the collection of event epochs into three splits within each subject: training set, validation set, and test set with a ratio of 3:1:1. The ratio between classes remained the same in each set. The loss function is categorical cross entropy (CCE) and the Adam optimizer is adopted with a learning rate of $10^{-3}$ and zero weight decay. The batch size is set to 32 and the total training epoch is 200.

## A.2 DESCRIPTION OF DATASETS

### A.2.1 DATASET 1: FEEDBACK ERROR-RELATED NEGATIVITY (ERN)

Error-related negativity (ERN) can be categorized as a kind of event-related potential (ERP), which occurs after an erroneous or abnormal event perceived by the subject. One characteristic of the feedback ERN is a relatively large negative amplitude approximately 350 ms and a positive amplitude approximately 500 ms after visual feedback triggered by the error event. In this work, we mainly use a well-studied EEG dataset from the BCI Challenge competition hosted by Kaggle to evaluate the artifact removal effectiveness. This dataset includes EEG recordings of 26 subjects (16 subjects labeled and 10 subjects unlabeled) that participated in a P300 speller task. P300 speller is a well-known BCI system that develops a typing application through P300 response evoked potential. The ERN experiment was conducted under the assumption that the ERN occurred if the subject received incorrect prediction (feedback) from the P300 speller. The objective of the competition was to improve the P300 speller performance by implementing error correction through ERN potentials. We used the 16 subjects with labeled data of which the sampling rate is 200 Hz initially with 56 passive Ag/AgCl EEG sensors.

In the interest of increasing the usability of EEG data, we applied some pre-processing procedures to each EEG recording. The EEG data were down-sampled to 128 Hz and re-referenced by the common average reference (CAR) method to eliminate common-mode noise and to zero-center the data. Each recording was band-pass filtered to 1-40 Hz through the FIR filter implemented by EEGLAB to remove DC drifting. During EEG decoding evaluation, we epoch EEG signals in [0, 1.25] second interval to obtain correct and erroneous feedback.

### A.2.2 DATASET 2: STEADY STATE VISUALLY EVOKED POTENTIAL (SSVEP)

Steady state visually evoked potential (SSVEP) is another kind of ERP that is characterized as periodic potential induced by rapidly repetitive visual stimulation. The SSVEP is composed of several discrete frequency components, which consist of the fundamental frequency of the visual stimulus as well as its harmonics. To investigate the generalization ability of the model, we use "EEG SSVEP Dataset II" from Multimedia Authoring & Management using your Eyes & Mind (MAMEM). The dataset includes EEG data from 11 subjects and consists of five different frequencies (6.66, 7.50, 8.57, 10.00, and 12.00 Hz). Each subject was recorded in five sessions and each session included 25 trials (5 trials for each class). The data used a 256-channel HydroCel Geodesic Sensor Net (HCGSN) and captured the signals with a sampling rate of 250 Hz. Since there are several different bad channels in each subject's EEG recording, we preserved 20 common channels from each subject to train the CLEEGN model. Every recording was down-sampled to 125 Hz, re-referenced by the common average reference (CAR) method, and band-pass filtered to 1-40 Hz. We epoch EEG signals in [1, 5] second interval for each event recorded timestamp. The first second was discarded under the consideration of a reaction delay of the subject.

### A.3 INVESTIGATION OF DECODING PERFORMANCE EVALUATION: SSVEP

In this work, we mainly use decoding performance as the assessment of different artifact removal methods. We provide an interpretation of the relationship between EEG quality and decoding performance. In Section 4 SSVEP result, IC-U-Net can optimize the weight with minimal MSE value among all compared artifact removal networks. However, the decoding performance is worse than CLEEGN and even the reference method (ICLabel), which implies that the average error over data points (MSE) is not the best assessment of EEG quality.

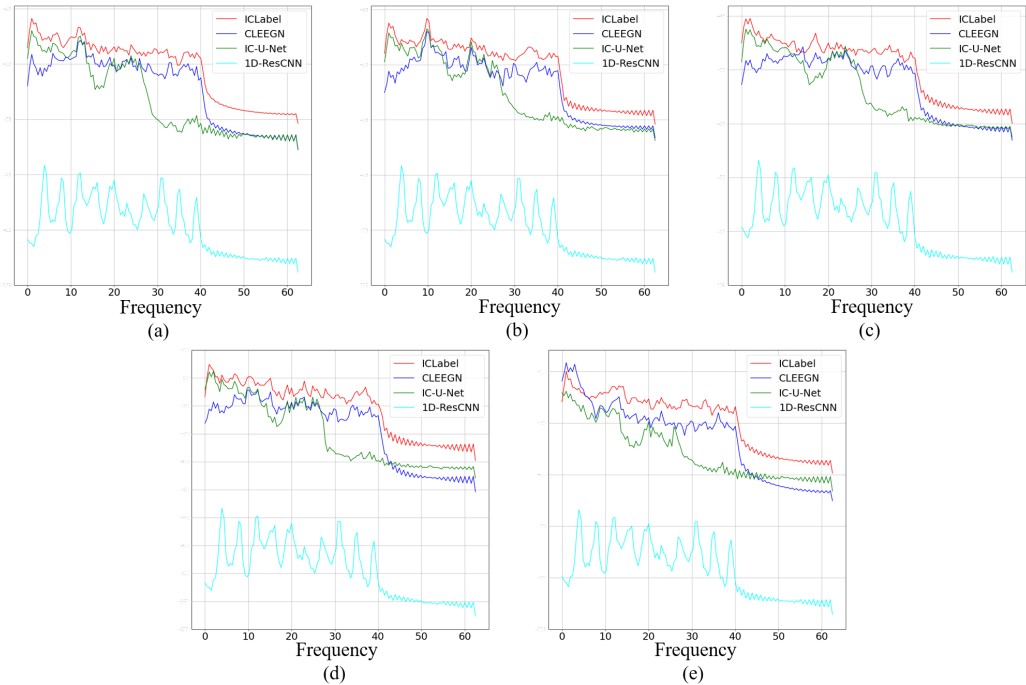

Figure 10: Power spectrum density of EEG data on different methods to each class

In the SSVEP experiment, the fundamental frequency of the external rapidly repetitive visual stimulation and its harmonics can be an important feature in classification. Hence, higher quality in the frequency component of EEG data is required. We use the power spectrum density (PSD) to interpret the result of decoding performance. Figure 10 shows the PSD of EEG event data in each class denoised by different methods. We can see the reconstructed EEG spectrum from CLEEGN in each class is similar to the reference. As for IC-U-Net, it seems that the model can not completely

reconstruct in several frequency bands (15-20Hz, 30-40Hz), which leads to the low decoding performance. The PSD result shows that there is great distortion in the power density of reconstructed data generated by 1D-ResCNN. Since 1D-ResCNN is a one-dimensional structure, we hypothesize that spatial information is important in EEG artifact removal.

## A.4 ADDITIONAL TABLES

| Method | ICLabel | | ASR-32 | | ASR-32-ICA | |
|---|---|---|---|---|---|---|
| Subject | MSE | AUC | MSE | AUC | MSE | AUC |
| S02 | 3.2835 | 0.8432 | 9.4341 | 0.8102 | 3.6285 | 0.8398 |
| S06 | 2.5723 | 0.6609 | 10.7221 | 0.6234 | 2.7348 | 0.6093 |
| S07 | 4.7804 | 0.7835 | 17.2428 | 0.7155 | 4.6049 | 0.6889 |
| S11 | 7.2317 | 0.6214 | 9.2276 | 0.5939 | 6.7069 | 0.6243 |
| S12 | 3.2090 | 0.7126 | 16.5107 | 0.6789 | 3.0982 | 0.6794 |
| S13 | 4.3666 | 0.7860 | 9.7730 | 0.7937 | 4.0034 | 0.7849 |
| S14 | 2.0056 | 0.8256 | 2.6510 | 0.7951 | 1.9174 | 0.8030 |
| S16 | 6.3664 | 0.6758 | 15.2617 | 0.6437 | 5.9677 | 0.6542 |
| S17 | 4.5280 | 0.8068 | 10.1852 | 0.8080 | 8.1861 | 0.8023 |
| S18 | 4.3724 | 0.7629 | 9.2606 | 0.6835 | 3.6407 | 0.7331 |
| S20 | 4.6405 | 0.7022 | 11.5464 | 0.6898 | 3.7116 | 0.6669 |
| S21 | 5.0383 | 0.7219 | 21.3344 | 0.6827 | 5.0986 | 0.6569 |
| S22 | 4.5625 | 0.7300 | 20.2808 | 0.6334 | 4.8429 | 0.6850 |
| S23 | 3.4311 | 0.7086 | 3.8892 | 0.6991 | 2.5044 | 0.7163 |
| S24 | 3.2064 | 0.7310 | 4.7346 | 0.7297 | 3.6940 | 0.7045 |
| S26 | 5.3556 | 0.5614 | 11.9691 | 0.5661 | 6.4727 | 0.5475 |

Table 4: Mean square error between reference denoising method and CLEEGN (MSE), AUC score of CLEEGN (AUC) in "BCI-Challenge" ERN dataset - Part1

| Method | ASR-16 | | ASR-16-ICA | | ASR-8 | |
|---|---|---|---|---|---|---|
| Subject | MSE | AUC | MSE | AUC | MSE | AUC |
| S02 | 4.2217 | 0.8205 | 3.5384 | 0.8425 | 4.0614 | 0.8205 |
| S06 | 4.4036 | 0.5513 | 3.2639 | 0.6388 | 3.9688 | 0.6388 |
| S07 | 11.4515 | 0.7116 | 5.1372 | 0.6588 | 8.2052 | 0.6717 |
| S11 | 8.4986 | 0.5803 | 4.9902 | 0.6061 | 7.0836 | 0.5983 |
| S12 | 7.1270 | 0.6980 | 4.1351 | 0.6690 | 4.6099 | 0.7005 |
| S13 | 5.0222 | 0.7870 | 6.5591 | 0.7680 | 4.5217 | 0.7778 |
| S14 | 2.2373 | 0.8137 | 2.5573 | 0.7723 | 2.6699 | 0.7982 |
| S16 | 9.6206 | 0.6680 | 4.9104 | 0.6554 | 6.8716 | 0.6660 |
| S17 | 18.8896 | 0.8076 | 10.9239 | 0.8041 | 27.6019 | 0.8141 |
| S18 | 5.2366 | 0.7139 | 3.6065 | 0.7264 | 4.3582 | 0.7006 |
| S20 | 6.7533 | 0.6843 | 5.5760 | 0.6668 | 4.3853 | 0.6776 |
| S21 | 8.5629 | 0.6501 | 6.6647 | 0.6338 | 6.4900 | 0.6674 |
| S22 | 9.3490 | 0.6395 | 5.1241 | 0.6460 | 6.7729 | 0.6419 |
| S23 | 3.1867 | 0.6795 | 4.4331 | 0.6874 | 2.4365 | 0.7022 |
| S24 | 2.7711 | 0.7191 | 3.7558 | 0.7130 | 3.7320 | 0.7117 |
| S26 | 7.6073 | 0.5861 | 6.4456 | 0.5648 | 9.1152 | 0.5652 |

Table 5: Mean square error between reference denoising method and CLEEGN (MSE), AUC score of CLEEGN (AUC) in "BCI-Challenge" ERN dataset - Part2

| Method | ASR-8-ICA | | ASR-4 | | ASR-4-ICA | |
|--------|-----------|-----------|---------|---------|-----------|---------|
| Subject | MSE | AUC | MSE | AUC | MSE | AUC |
| S02 | 3.4726 | 0.8277 | 4.0551 | 0.8204 | 3.3401 | 0.8249 |
| S06 | 2.9740 | 0.6257 | 4.4043 | 0.5893 | 4.1949 | 0.6526 |
| S07 | 5.4224 | 0.6423 | 8.4744 | 0.6592 | 4.7362 | 0.6969 |
| S11 | 4.3832 | 0.6035 | 9.2528 | 0.5868 | 4.9227 | 0.5972 |
| S12 | 3.5382 | 0.6800 | 3.9801 | 0.6969 | 3.8259 | 0.6732 |
| S13 | 5.4227 | 0.7921 | 5.5527 | 0.7833 | 6.5429 | 0.7628 |
| S14 | 2.1820 | 0.8004 | 3.6646 | 0.7947 | 2.9259 | 0.8057 |
| S16 | 4.8361 | 0.6685 | 8.4332 | 0.6379 | 4.0452 | 0.6757 |
| S17 | 11.1061 | 0.8014 | 24.9330 | 0.8102 | 16.4403 | 0.8141 |
| S18 | 3.4891 | 0.7191 | 4.0793 | 0.7468 | 3.2979 | 0.7322 |
| S20 | 4.6687 | 0.6625 | 4.9512 | 0.6444 | 4.6338 | 0.6505 |
| S21 | 6.1690 | 0.6518 | 5.0372 | 0.6479 | 4.5595 | 0.6507 |
| S22 | 5.1410 | 0.6399 | 6.7985 | 0.6566 | 5.5497 | 0.6801 |
| S23 | 3.5369 | 0.6935 | 3.6311 | 0.6993 | 3.9354 | 0.6870 |
| S24 | 2.8299 | 0.7215 | 4.2418 | 0.7231 | 3.7892 | 0.7075 |
| S26 | 6.8030 | 0.5495 | 12.7965 | 0.5704 | 7.7062 | 0.5595 |

Table 6: Mean square error between reference denoising method and CLEEGN (MSE), AUC score of CLEEGN (AUC) in "BCI-Challenge" ERN dataset - Part3

| DataLength (min) | ICLabel | 30 | | 20 | | 10 | |
|------------------|---------|---------|---------|---------|---------|---------|---------|
| Subject | AUC | MSE | AUC | MSE | AUC | MSE | AUC |
| S02 | 0.8114 | 3.4818 | 0.8290 | 3.6152 | 0.8359 | 3.5139 | 0.8191 |
| S06 | 0.6281 | 2.4449 | 0.6117 | 2.4496 | 0.6195 | 2.3353 | 0.6215 |
| S07 | 0.7298 | 4.1740 | 0.7177 | 4.3742 | 0.6744 | 4.2118 | 0.7166 |
| S11 | 0.5971 | 7.0916 | 0.6050 | 6.8226 | 0.6107 | 7.6580 | 0.6037 |
| S12 | 0.6916 | 2.6238 | 0.6700 | 2.7464 | 0.6724 | 2.3652 | 0.6949 |
| S13 | 0.7859 | 3.7257 | 0.7855 | 3.9133 | 0.7774 | 3.5441 | 0.7926 |
| S14 | 0.7888 | 1.7715 | 0.7766 | 1.7445 | 0.7959 | 1.6376 | 0.8086 |
| S16 | 0.6503 | 5.5278 | 0.6585 | 5.5853 | 0.6811 | 5.2618 | 0.6878 |
| S17 | 0.8198 | 4.6574 | 0.8162 | 4.4208 | 0.8069 | 4.4521 | 0.8066 |
| S18 | 0.6997 | 3.3821 | 0.6965 | 3.4677 | 0.6899 | 3.7442 | 0.7045 |
| S20 | 0.6829 | 3.4097 | 0.6873 | 3.8692 | 0.6669 | 3.4116 | 0.6960 |
| S21 | 0.6198 | 4.0377 | 0.6760 | 3.7572 | 0.6918 | 3.7251 | 0.6733 |
| S22 | 0.6372 | 3.2582 | 0.6920 | 3.4192 | 0.6383 | 3.0667 | 0.6600 |
| S23 | 0.6985 | 2.3770 | 0.6994 | 2.6990 | 0.7050 | 2.3144 | 0.7213 |
| S24 | 0.7339 | 2.1982 | 0.7108 | 2.1782 | 0.7317 | 2.0520 | 0.7396 |
| S26 | 0.5867 | 3.9663 | 0.5701 | 3.9898 | 0.5524 | 4.0011 | 0.5566 |

Table 7: Mean square error between reference denoising method and CLEEGN (MSE), AUC score of CLEEGN (AUC) in "BCI-Challenge" ERN dataset trained by different data length - Part1

| DataLength (min) | ICLabel | 4 | | 2 | | 1 | |
|---|---|---|---|---|---|---|---|
| Subject | AUC | MSE | AUC | MSE | AUC | MSE | AUC |
| S02 | 0.8114 | 3.2609 | 0.8101 | 3.1745 | 0.8324 | 3.3467 | 0.8213 |
| S06 | 0.6281 | 2.2442 | 0.6093 | 3.0822 | 0.6135 | 3.3718 | 0.6302 |
| S07 | 0.7298 | 4.2698 | 0.6655 | 4.6784 | 0.6774 | 3.8478 | 0.7180 |
| S11 | 0.5971 | 6.5970 | 0.6135 | 7.6816 | 0.6406 | 4.5129 | 0.6193 |
| S12 | 0.6916 | 2.7294 | 0.6948 | 2.7830 | 0.6733 | 3.7798 | 0.6717 |
| S13 | 0.7859 | 3.4267 | 0.7793 | 5.7493 | 0.8007 | 6.3488 | 0.7760 |
| S14 | 0.7888 | 1.6493 | 0.7934 | 2.2690 | 0.7937 | 2.5422 | 0.7980 |
| S16 | 0.6503 | 5.9354 | 0.6681 | 7.9410 | 0.6594 | 5.2486 | 0.6584 |
| S17 | 0.8198 | 4.5170 | 0.7939 | 4.7344 | 0.7965 | 5.8205 | 0.7986 |
| S18 | 0.6997 | 3.8781 | 0.7546 | 4.6024 | 0.7202 | 4.7593 | 0.7239 |
| S20 | 0.6829 | 4.1724 | 0.6721 | 4.3053 | 0.6837 | 5.9014 | 0.6707 |
| S21 | 0.6198 | 4.5480 | 0.6905 | 4.8047 | 0.6494 | 5.2452 | 0.6497 |
| S22 | 0.6372 | 3.3430 | 0.6629 | 4.0498 | 0.6772 | 5.7725 | 0.6739 |
| S23 | 0.6985 | 2.9178 | 0.6939 | 3.1362 | 0.6936 | 4.3290 | 0.6796 |
| S24 | 0.7339 | 2.3747 | 0.7173 | 2.8420 | 0.7468 | 4.5555 | 0.7275 |
| S26 | 0.5867 | 4.2618 | 0.5666 | 4.9736 | 0.5705 | 6.8117 | 0.5682 |

Table 8: Mean square error between reference denoising method and CLEEGN (MSE), AUC score of CLEEGN (AUC) in "BCI-Challenge" ERN dataset trained by different data length - Part2

| Method | ICLabel | | CLEEGN | | IC-U-Net | | 1D-ResCNN | |
|---|---|---|---|---|---|---|---|---|
| Subject | MSE | AUC | MSE | AUC | MSE | AUC | MSE | AUC |
| S02 | - | 0.8238 | 3.5139 | 0.8274 | 4.4294 | 0.7816 | 5.7621 | 0.7598 |
| S06 | - | 0.6616 | 2.3353 | 0.6926 | 4.1464 | 0.6132 | 5.4486 | 0.5601 |
| S07 | - | 0.8263 | 4.2118 | 0.7410 | 4.3550 | 0.7003 | 6.5909 | 0.6848 |
| S11 | - | 0.6083 | 7.6580 | 0.6324 | 5.0279 | 0.6153 | 5.1880 | 0.5729 |
| S12 | - | 0.6947 | 2.3652 | 0.6756 | 3.8519 | 0.6587 | 7.0038 | 0.6283 |
| S13 | - | 0.7913 | 3.5441 | 0.7835 | 8.1448 | 0.7469 | 8.1590 | 0.7509 |
| S14 | - | 0.8128 | 1.6376 | 0.8135 | 2.9057 | 0.7674 | 3.7053 | 0.7679 |
| S16 | - | 0.6402 | 5.2618 | 0.6615 | 4.0351 | 0.6405 | 6.1420 | 0.6485 |
| S17 | - | 0.8177 | 4.4521 | 0.8153 | 8.2195 | 0.7612 | 7.8305 | 0.7564 |
| S18 | - | 0.7419 | 3.7442 | 0.7667 | 4.7798 | 0.7082 | 6.4393 | 0.6729 |
| S20 | - | 0.6962 | 3.4116 | 0.6770 | 5.7497 | 0.6802 | 8.9663 | 0.6359 |
| S21 | - | 0.7130 | 3.7251 | 0.7742 | 5.1049 | 0.7132 | 7.4864 | 0.6718 |
| S22 | - | 0.7166 | 3.0667 | 0.7435 | 6.1630 | 0.7199 | 10.6583 | 0.6749 |
| S23 | - | 0.7016 | 2.3144 | 0.7316 | 4.3579 | 0.6848 | 3.6919 | 0.6524 |
| S24 | - | 0.7366 | 2.0520 | 0.7228 | 3.7706 | 0.7186 | 4.7246 | 0.7245 |
| S26 | - | 0.5663 | 4.0011 | 0.5445 | 8.2955 | 0.6077 | 9.6387 | 0.5539 |

Table 9: ERN mean square error and AUC score of each subject on different network structure

| Method | ICLabel | | CLEEGN | | IC-U-Net | | 1D-ResCNN | |
| Subject | MSE | Top-1 Acc | MSE | Top-1 Acc | MSE | Top-1 Acc | MSE | Top-1 Acc |
|---|---|---|---|---|---|---|---|---|
| S001 | - | 0.6930 | 0.0859 | 0.6620 | 0.0852 | 0.4570 | 0.1228 | 0.2335 |
| S002 | - | 0.7350 | 0.1379 | 0.8000 | 0.0609 | 0.3235 | 0.0608 | 0.2315 |
| S003 | - | 0.3175 | 0.1113 | 0.3645 | 0.1013 | 0.2830 | 0.1798 | 0.2325 |
| S004 | - | 0.2610 | 0.1040 | 0.2770 | 0.0984 | 0.2380 | 0.1896 | 0.2330 |
| S005 | - | 0.2870 | 0.0387 | 0.2985 | 0.0475 | 0.2685 | 0.0539 | 0.2310 |
| S006 | - | 0.5300 | 0.0237 | 0.5595 | 0.0322 | 0.3910 | 0.0583 | 0.2290 |
| S007 | - | 0.5110 | 0.0819 | 0.4935 | 0.0768 | 0.4035 | 0.1503 | 0.2340 |
| S008 | - | 0.2020 | 0.1143 | 0.2270 | 0.1169 | 0.2015 | 0.1205 | 0.2345 |
| S009 | - | 0.9050 | 0.0492 | 0.8005 | 0.0460 | 0.4070 | 0.3755 | 0.4070 |
| S010 | - | 0.5560 | 0.0405 | 0.5055 | 0.0379 | 0.3690 | 0.3125 | 0.5260 |
| S011 | - | 0.4360 | 0.0668 | 0.6870 | 0.0651 | 0.4225 | 0.6281 | 0.4075 |

Table 10: SSVEP mean square error and top-1 accuracy of each subject on different network structure

