# OpenReview forum: "CLEEGN: A Convolutional Neural Network for Plug-and-Play Automatic EEG Reconstruction"
_ICLR.cc/2023/Conference — Submitted to ICLR 2023_

### Official Review · Reviewer_fmQB · 2022-10-25

**Confidence:** 5
**Correctness:** 2
**Technical Novelty And Significance:** 1
**Empirical Novelty And Significance:** 1
**Recommendation:** 3

**Clarity, Quality, Novelty And Reproducibility:**

Unfortunately, the manuscript lacks details on reproducing the machine-learning model since only a colorful model picture is available.

**Strength And Weaknesses:**

The application of deep learning to EEG denoising looks interesting, but novelty in the proposed approach is lacking. The authors seem aware of ICA and EEGNet, but other successful strategies, such as EMD or BrainDecode are unfortunately ignored. Another submission limitation is a minimal and not-that-perfect evaluation on a tiny open-access dataset. The proposed CLEEGN is a standard and of the shelf available convolutional encoded + decoder, thus hard to find novelty in the proposed approach.

**Summary Of The Paper:**

The authors present an approach to utilize a convolutional neural network (CLEEGN) with limited novelty (there are so many such approaches on the market already) and super limited evaluation on a tiny EEG dataset. Due to little originality and minimal assessment, the paper is not suitable for the ICLR.

**Summary Of The Review:**

The authors, unfortunately, submitted a very preliminary work lacking technical, novelty, and implementation details. A limited evaluation and small classification gain further support for a vague idea evaluation. At ICLR, more novel and well-supported results interest the audience.

---

> ### Author Response · Authors · 2022-11-18
> **Response to Reviewer fmQB**
>
> (1) The application of deep learning to EEG denoising looks interesting, but novelty in the proposed approach is lacking.
>
> Reply: We agree with the reviewer that the novelty of an original research work is important. This work proposes the first deep learning model dedicated for real-time and plug-and-play EEG reconstruction. The customized design takes the characteristics of EEG data into account and results in high quality EEG reconstruction that outperforms all other DL-based methods and even the conventional offline, calibration-needed methods according to the objective assessment of classification accuracy. We have addressed the novelty of this breakthrough in Section 1 and 5.
>
> (2) The authors seem aware of ICA and EEGNet, but other successful strategies, such as EMD or BrainDecode are unfortunately ignored.
>
> Reply: We thank the authors for suggesting additional relevant techniques that potentially can be included in our framework. We would definitely seek applicable automatic EMD-based artifact removal method for preparing noiseless reference EEG data for training our model in the future. Meanwhile, the Shallow/Deep ConvNet available in the BrainDecode and many other end-to-end EEG classifiers can serve as alternative classifiers to measure the EEG quality. We have included discussion about these future works in Section 5.

---

> ### Author Response · Authors · 2022-12-12
> **Looking forward to you response**
>
> Thank you for your inspiring questions about our work. We have updated the manuscript with additional discussion accordingly. We appreciate all the comments and look forward to further discussion.

---

### Official Review · Reviewer_8xpw · 2022-10-25

**Confidence:** 4
**Correctness:** 3
**Technical Novelty And Significance:** 2
**Empirical Novelty And Significance:** 3
**Recommendation:** 5

**Clarity, Quality, Novelty And Reproducibility:**

The paper is well-organized but the presentation has minor details that could be improved. The paper appears to be technically sound. The novelty of the paper is relatively ordinary. The implementation code is not provided reducing the reproducibility.

**Strength And Weaknesses:**

Strength:
1.A lightweight autoencoder CNN is proposed for EEG reconstruction.
2.The model can reconstruct subject-independent EEG without any training/calibration for a new subject.
3.The model outperforms leading methods in providing reconstructed EEG data with the optimal decoding performance for some BCI datasets.

Weaknesses:
1. In Figure 1, the differentiation of each part in the encoder and decoder is not clear and it is recommended that it be redrawn.
2. How can the Decoder part of the model process data of size [1*T*C] and output a result of size [C*T*1] using a convolution kernel of size (C,1)?
3. Is the purpose of denoising and reconstructing to approximate the effect of the offline denoising method? The reference EEG is demanding and how to ensure that the reference EEG is clean.
4. There are certain errors in the Decoder module in Table 1.
5. The experimental setup is unclear. How are the data sets divided? Is the data used after some pre-processing, or is it raw data without any processing? How are the model parameters set?
6. The experimental results of EEGNet on the raw data should be added to Tables 2 and 3.

**Summary Of The Paper:**

In this paper, the authors proposed a novel convolutional neural network (CLEEGN) for plug-and-play automatic EEG reconstruction. CLEEGN can reconstruct subject-independent EEG without any training/calibration for a new subject. The performance of CLEEGN was validated using multiple evaluations including reconstruction error assessment, decoding accuracy, etc. In general, the paper is well-written and easy to understand.

**Summary Of The Review:**

The innovation of this work is general. The proposed model aims to achieve online EEG artifact removal and reconstruction. A lightweight autoencoder-based CNN was used and performance was verified by a variety of evaluations including waveform observation, reconstruction error evaluation, and decoding accuracy. The paper is likely to have a modest impact within a subfield of AI.

---

> ### Author Response · Authors · 2022-11-18
> **Response to Reviewer 8xpw**
>
> (1) 1. A lightweight autoencoder CNN is proposed for EEG reconstruction. 2.The model can reconstruct subject-independent EEG without any training/calibration for a new subject. 3.The model outperforms leading methods in providing reconstructed EEG data with the optimal decoding performance for some BCI datasets.
>
> Reply: Thank you for your valuable comments that helped us improve this article.
>
> (2) In Figure 1, the differentiation of each part in the encoder and decoder is not clear and it is recommended that it be redrawn.
>
> Reply: Thanks you for the suggestion. We apologize for the errors in the figure and the table. We have corrected the annotations and fixed the errors. Please refer to the modified Section 3, Figure 1 and Table 1 highlighted in red.
>
> (3) How can the Decoder part of the model process data of size [1TC] and output a result of size [CT1] using a convolution kernel of size (C,1)? There are certain errors in the Decoder module in Table 1.
>
> Reply: We apologize the mismatching between the Figure 1 and the Table 1. The annotation of the output shape of fourth convolutional layer has been carefully reviewed and fixed. The correct shape is in the Table 1. This was due to the misplacement of the channel-last expression to the channel-first version. To avoid further confusion, we updated both Figure 1 and Table 1 using the channel-first expression that matches the model structure in our source code.
>
> (4) Is the purpose of denoising and reconstructing to approximate the effect of the offline denoising method? The reference EEG is demanding and how to ensure that the reference EEG is clean.
>
> Reply: We thank the reviewer for the important questions. The proposed CLEEGN model learns about the processing of denoising and reconstruction through the paired noisy/reference EEG data generated by an existing offline denoising method. To our best knowledge, there is no well-defined true cleanness of EEG data, and thus neither there exists a method to determine if a set of EEG data is clean. Therefore, we herein employed the classification accuracy to measure the quality of EEG data, and identify that the ICA+ICLabel is our best automatic approach to generate large amount of training data.
>
> (5) The experimental setup is unclear. How are the data sets divided? Is the data used after some pre-processing, or is it raw data without any processing? How are the model parameters set?
>
> Reply: We thank the reviewer for pointing out the insufficiency in the description. Since we intend to train a subject-independent model for artifact removal without any calibration. We use the leave-k-subjects-out training scheme, which means that the subjects for test are strictly excluded in the training/validation set. The splitting strategy of training and validation are described as in Section 4 along with the information such as data length per subject and number of subjects. Due to the page limitation, we provide the description of ERN "BCI-Challenge" dataset and SSVEP "MAMEM-SSVEP-II" dataset in A.2 DESCRIPTION OF DATASETS in the Appendix. The general data pre-processing methods on EEG we adopted are also provided in this paragraph. The model parameters configuration are provided in A.1 NEURAL NETWORK TRAINING CONFIGURATION. In this section, we provide the model configuration of our CLEEGN model and each reference model. The setting of the classification model, EEGNet, is also provided in this paragraph.
>
> (6) The experimental results of EEGNet on the raw data should be added to Tables 2 and 3.
>
> Reply: We appreciate this suggestion and have updated Table 2 and 3 with the classification performance of EEGNet using raw EEG data. This result suggests the importance of artifact removal for elevating EEG data quality.
>
> (7) The implementation code is not provided reducing the reproducibility.
>
> Reply: The codes are available in the supplementary material with our first submission. We have added additional instruction in the code appendix, re-organized the supplementary material, and provided sample data with pre-trained model weights for a fast replication of our results.

---

> ### Author Response · Authors · 2022-12-12
> **Looking forward to your response**
>
> Thank you for your valuable comments. We have revised the manuscript thoroughly with additional experimental results and discussion, as detailed in our response earlier. We truly appreciate your comments and look forward to your further response.

---

### Official Review · Reviewer_sQXi · 2022-10-25

**Confidence:** 3
**Correctness:** 4
**Technical Novelty And Significance:** 3
**Empirical Novelty And Significance:** 3
**Recommendation:** 8

**Clarity, Quality, Novelty And Reproducibility:**

The manuscript clearly describes an idea that seems novel to me, and availability of code should also ease reproducibility.

**Strength And Weaknesses:**

**Update**

Thanks for adding relevant discussion points to the paper, good that it is at least addressed a bit.


**Pre-Rebuttal**

The idea is straightforward, clearly explained and clearly evaluated.

One open question to me is how far the training setup of using the results of offline artifact removal as the ground truth limits the potential performance of the overall method. Could the authors also envision instead synthetically adding artifacts to clean EEG (potentially also synthetic) data? Not necessarily as part of this manuscript, but these questions would be worth discussing a bit.

Also more datasets and cross-dataset performance may be interesting, like would it be possible to go towards a general-purpose reconstruction method that can be applied to a new dataset without training? Some discussion on that could be interesting as well.



**Summary Of The Paper:**

The work proposes an online deep-learning encoding-decoding model for EEG denoising/artifact removal. The work uses established offline artifact removal methods to create training data, so the data after offline artifact removal is considered the reference ground truth data that should be reconstructed from the noisy EEG signal. The reconstruction results are evaluated by mean squared error to the reference data and by the decoding accuracies achieved on the reconstructed/cleaned data.

**Summary Of The Review:**

This seems a clear straightforward method with a clear evaluation, further evaluation on more data may increase the scientific value of the manuscript further.

---

> ### Author Response · Authors · 2022-11-18
> **Response to Reviewer sQXi**
>
> Thank you for your valuable comments that helped us improve this study. We have responded to all of the comments and questions as described below.
>
> (1) One open question to me is how far the training setup of using the results of offline artifact removal as the ground truth limits the potential performance of the overall method. Could the authors also envision instead synthetically adding artifacts to clean EEG (potentially also synthetic) data? Not necessarily as part of this manuscript, but these questions would be worth discussing a bit.
>
> Reply: We highly appreciate this open question about the training setup. Regarding using synthetic data to train a artifact removal method, there are at least two major issues should be considered. First, the synthetic process consists of pre-defined artifact waveforms (e.g. EOG, EMG, EKG artifacts, line noise, etc) and mixing approach. This limits the diversity of the artifacts/noise in the synthetic data into several well-known types of artifacts that can be prepared in advance. Meanwhile, the mixing approach is often linear and thus simplify the unmixing problem to an unrealistic level. The other issue is the validation of EEG cleanness. To our best knowledge, there is no well-defined true cleanness of EEG data, and thus neither is there exist a method to determine whether a set of EEG data is clean or not. We cannot say the reference EEG data sets are perfectly clean, but are definitely more noiseless according to the objective assessment using classification accuracy. This allows us to evaluate the proposed model using real EEG data where various artifacts/noise present naturally. We have addressed this aspect in Section 3.
>
> (2) Also more datasets and cross-dataset performance may be interesting, like would it be possible to go towards a general-purpose reconstruction method that can be applied to a new dataset without training? Some discussion on that could be interesting as well.
>
> Reply: We appreciate the reviewer for the suggestions. The type of EEG patterns does not affect the operation of our framework, and therefore this is absolutely an interesting extension in the future. Another critical future work is the cross-montage learning of our model, as it currently can inference only on the data with the same montage. We have added the discussion in Section 5.
>
> (3) Availability of code should also ease reproducibility.
>
> Reply: We appreciate this suggestion and have re-organized the supplementary material and included a small set of data samples and pre-trained model weights for quick reproduction of our work.

---

> > ### Comment · Reviewer_sQXi · 2022-11-23
> > **Thanks for your answers.**
> >
> > Thanks for adding relevant discussion points to the paper, good that it is at least addressed a bit, increased the score.

---

> > > ### Author Response · Authors · 2022-11-28
> > > **Thanks for your positive response.**
> > >
> > > We truly appreciate your positive response and score increase.

---

### Official Review · Reviewer_KYwV · 2022-10-29

**Confidence:** 3
**Clarity, Quality, Novelty And Reproducibility:** The write-up is clear. The work is no…
**Correctness:** 3
**Technical Novelty And Significance:** 2
**Empirical Novelty And Significance:** 2
**Recommendation:** 5

**Strength And Weaknesses:**

Strengths:
- Performs well according to reported results

Weaknesses:
- The method is fairly simple simple based a CNN with mirrorred layers that are referred to as encoder and decoder
- There are just two baselines and its surprising that such a simple model can outperform both references. Perhaps enough algorithmic work has not been done in the domain.
- The reference models are also based on very old architecture, i.e. U-Net and ResNet.
- The interpretability aspect makes little sense where PCA is used to project raw signals into a 2-D space and the relative positions are said to be similar.

**Summary Of The Paper:**

The paper proposes a method for EEG reconstruction based on CNN.

**Summary Of The Review:**

The paper proposes a method that is fairly simple simple based a very simple CNN. The lack of good baselines and weak interpretability claim make this paper hard to recommend.

---

> ### Author Response · Authors · 2022-11-18
> **Response to Reviewer KYwV**
>
> (1) Performs well according to reported results
>
> Reply: We thank the review for this positive comment.
>
> (2) The method is fairly simple simple based a CNN with mirrorred layers that are referred to as encoder and decoder
>
> Reply: We acknowledge the concern about the simplicity of the model structure. We design the proposed CLEEGN model for the purpose of real-time EEG reconstruction, and thus the simplicity of model is highly favorable as it brings computational efficiency. Considering the superiority in the performance EEG reconstruction assessed by the visual inspection and increased classification accuracy, the simple architecture of our model is regarded as an important criterion for real-time processing.
>
> (3) There are just two baselines and its surprising that such a simple model can outperform both references. Perhaps enough algorithmic work has not been done in the domain. The reference models are also based on very old architecture, i.e. U-Net and ResNet.
>
> Reply: Thank you for pointing out the scarcity of the baseline methods for comparison. There is still lack of development in the field of neural-network-based EEG artifact removal and reconstruction. To the best of our knowledge, very few related works have provided source codes properly for reproduction. We have worked on our best to seek suitable baseline methods but only found two additional methods, relatively out-dated, into the comparison shown in Table 2 and 3.
>
> (4) The interpretability aspect makes little sense where PCA is used to project raw signals into a 2-D space and the relative positions are said to be similar.
>
> Reply: We agree with the review that the PCA can project raw signals into a 2-D space with spatial relationship of multi-channel EEG is retained. We leverage PCA to visualize noisy EEG, latent features, and reconstructed EEG all in the same domain as a thorough comparison. This allow us to observe the process of reconstruction in a high-dimensional latent space. As observed in the PCA projection, adjacent EEG channels tend to retain their relative spatial relationship as nearby dots in the PCA space.
>
> (5) Code is not available.
>
> Reply: The codes are available in the supplementary material with our first submission. We have added additional instruction in the code appendix, re-organized the supplementary material, and provided sample data with pre-trained model weights for a fast replication of our results.

---

> > ### Comment · Reviewer_KYwV · 2022-11-23
> > **Thanks for the response**
> >
> > Thanks for the response. I apologize for missing the submitted code. In terms of the method and its novelty, I still hold the same opinion. Missing codes should not be the reason that other baselines are not included. The same dataset can be used with the same training/testing conditions for the proposed model so it can be compared. I still think the baselines are weak and the method does not exude enough alignment with recent developments to suggest that it is an improvement from current state-of-the-art. So, I will hold my original score.

---

> > > ### Author Response · Authors · 2022-11-28
> > > **Thanks for your response.**
> > >
> > > We are sorry that our response was not satisfactory enough. It is unfortunate that the baseline methods included for comparison in this study are considered weak for you. In fact, most of them are able to perform effective artifact removal for EEG signals, and thus it is to our delight that our new method can outperform those counterparts with a simpler structure. It is worth noting that leading deep-learning-based models for EEG processing are mostly not as complex as those for other domains e.g. image recognition. It is unlikely that we failed to include the leading techniques of EEG reconstruction in this study, but still we would truly appreciate if the reviewer could point out any other recently developed SOTA methods for us to include for comparison. Again, we thank for your time and suggestions that have already helped us improve the quality of this paper.

---

### Decision · Program_Chairs · 2023-01-20

**Decision:**

Reject

**Justification For Why Not Higher Score:**

The technical novelty is limited.  The baselines are weak and the method does not exude enough alignment with recent developments to suggest that it is an improvement from current state-of-the-art.

**Justification For Why Not Lower Score:**

N/A

**Metareview: Summary, Strengths And Weaknesses:**

Summary: The work proposes an online deep-learning encoding-decoding model for EEG denoising/artifact removal. CLEEGN can reconstruct subject-independent EEG without any training/calibration for a new subject. The performance of CLEEGN was validated using multiple evaluations including reconstruction error assessment, decoding accuracy, etc. In general, the paper is well-written and easy to understand.

Strengths: The application of deep learning to EEG denoising is interesting. The model outperforms leading methods in providing reconstructed EEG data with the optimal decoding performance for some BCI datasets.

Novelty: The technical novelty is limited and the proposed CLEEGN is a standard and of the shelf available convolutional encoded + decoder. The interpretability aspect makes little sense where PCA is used to project raw signals into a 2-D space and the relative positions are said to be similar.